# Increased Preeclampsia Risk in GDM Pregnancies: The Role of SIRT1 rs12778366 Polymorphism and Telomere Length

**DOI:** 10.3390/ijms26072967

**Published:** 2025-03-25

**Authors:** Olga Dmitrenko, Nataliia Karpova, Malik Nurbekov

**Affiliations:** Federal State Budgetary Institution “Research Institute of Pathology and Pathophysiology”, 125315 Moscow, Russia; dolga6528@gmail.com (O.D.); mlkn0361@xmail.ru (M.N.)

**Keywords:** gestational diabetes mellitus, preeclampsia, polymorphism, SIRT1, rs12778366, rs7895833, relative telomere length, telomere length

## Abstract

Preeclampsia (PE) and gestational diabetes mellitus (GDM) are common pregnancy disorders with shared pathophysiological mechanisms. This study examined the association between SIRT1 polymorphisms (rs12778366 and rs7895833) and relative telomere length (RTL) in women with PE and GDM. The DNA from pregnant women with GDM with and without PE was analyzed. The RTL and genotyping were measured using quantitative real-time PCR. The women with GDM and PE had significantly shorter telomeres. The rs12778366 TC genotype was associated with a 4.48-fold increased risk of PE (OR = 4.48; 95% CI 1.54–13.08; *p* = 0.003). The PE group had a higher prevalence of the heterozygous TC rs12778366 genotype with short telomeres. The SIRT1 variant rs12778366 is associated with shorter telomeres and an increased risk of developing preeclampsia, suggesting it may be a useful biomarker for preeclampsia risk assessment in GDM pregnancies.

## 1. Introduction

Preeclampsia (PE) and gestational diabetes mellitus (GDM) are common pregnancy complications with similar risk factors (maternal age ≥35 years, multiple pregnancies, and body mass index ≥ 30 kg/m^2^) and similar pathophysiological mechanisms [1,2,3]. GDM is diabetes that is first diagnosed in the second or third trimester of pregnancy that is not clearly either preexisting type 1 or type 2 diabetes [4]. It is characterized by the relative inability of pancreatic β-cells to sufficiently respond to the increased need for insulin during pregnancy, which contributes to the development of hyperglycemia of varying degrees [5]. Data from various years point to a much greater preeclampsia rate in the case of gestational diabetes mellitus (7.3%) vs. the general population (4.5%) [6,7,8,9,10]. PE is a multi-system disorder that occurs after gestational week 20 and is characterized by arterial hypertension combined with proteinuria (≥0.3 g/L in daily urine), edema, and multi-organ dysfunction [11,12,13]. GDM complicated by PE increases the risk of perinatal adverse events with a greater impact on the future health of the mother and the offspring.

The data from several studies suggest a connection between shortened telomere length and an increased risk of pregnancy-related complications, including PE [14]. During pregnancy, the placental cells undergo morphological changes that indicate aging [15,16,17,18]. Telomeres play a key role in the process of cellular aging [6,18,19,20]. Telomeres are repeated nucleotide structures (TTAGGG) located at the ends of the eukaryotic linear chromosomes that protect the terminal regions of chromosomal DNA from degradation [21]. An incomplete chromosome replication during cell division and harmful agents like nucleases, oxidative processes, and free oxygen radicals can damage telomeres.

Many studies have shown that the length of leukocyte telomeres is an inherited trait, and estimates of heritability range from 34 to 82% [22]. Single nucleotide polymorphisms (SNPs) are inherited from parents and transmit inherited events [23]. Several candidate genes have been identified, such as TERT, TERC, OBFC1, ZNF676, OBFC1, CTC1, and SIRT1, which are associated with interindividual variation in telomere length [24,25,26,27,28]. Kim et al. (2012) showed that SNPs located in the SIRT1 gene have a significant effect on telomere length [28]. NAD+-dependent deacetylase SIRT1 ensures telomere homeostasis by inducing telomerase and shelterin protein expression and regulating telomere heterochromatin formation [29]. The decrease in NAD+ levels with age contributes to the inactivation of SIRT1, which leads to the shortening of telomeres and the genetic instability of cells [30,31]. The results of a study by Pieters et al. (2015) demonstrated a positive correlation of SIRT1 gene expression with the length of leukocyte telomeres [32]. In turn, telomere dysfunction leads to the suppression of SIRT1 expression [33].

Expression of the SIRT1 gene affects the performance of the SIRT1 protein in diverse tissues and organs, such as the endothelial cells of blood vessels and the placenta [34,35,36]. The SIRT1 gene is located in the 10th chromosome in the q21.3 locus and comprises 11 exons. Previous studies have shown that SNPs are more common in the promoter regions than in the coding regions of a gene and can affect its expression [37]. The polymorphic variants rs12778366 and rs7895833 in the SIRT1 gene promoter region, according to few studies, are associated with age-related macular degeneration (AMD), type 2 diabetes mellitus (T2DM), obesity, glucose tolerance, the risk of hypertension, and PE [38,39,40,41,42,43,44].

Because an individual’s genotype remains largely unchanged throughout life and serves as an objective indicator unaffected by environmental factors, and considering telomeres as potential biomarkers of preeclampsia, we hypothesized a correlation between telomere length and the SIRT1 gene polymorphisms rs12778366 and rs7895833 in pregnant women with GDM. This study aimed to investigate the correlation between telomere length and these selected SNPs, with the goal of potentially providing earlier detection of a genetic risk for preeclampsia in GDM pregnancies.

## 2. Results

Genotyping and the relative telomere length were analyzed in 61 patients with a combination of GDM and PE (PE group) and 63 pregnant women with GDM without PE (control group).

Hardy–Weinberg equilibrium (HWE) analysis in the control group showed that the genotype distribution of rs7895833 was consistent with HWE (*p* = 0.05). The genotype distribution of rs12778366 approached, but did not reach, statistical significance for deviation from HWE (*p* > 0.05) (Table 1). We did not exclude this variant from the analysis for several reasons. Deviation from HWE can arise from factors such as natural selection, inbreeding, population stratification, copy number variation, or genotyping errors [45,46]

Genotyping errors were eliminated by the repeated genotyping of 30% of the samples, which were randomly selected. Another probable reason for the deviation of the observed genotype frequencies for the rs12778366 polymorphism in this group is the low frequency of the minor C allele. According to the NCBI database (ALFA project), the minor allele frequency (MAF) of the C allele for rs12778366 is 0.00227 (sample size = 11,430). This low MAF is consistent with our observed genotype distribution, given our sample size.

Association analysis revealed a significant association between the rs12778366 polymorphism of the SIRT1 gene and PE in women with GDM. Specifically, the individuals with the heterozygous TC genotype showed a 4.48-fold increased risk of developing PE under a dominant inheritance model (*p* = 0.003) (Table 2).

The mean age of the pregnant women in the PE group was 32.31 years (SD = 4.68), and in the group without preeclampsia was 30.78 years (SD = 4.72) (Figure 1). There were no significant differences in age between the study groups (*p*-value > 0.05).

The mean telomere length in the PE group was 0.89 (SD = 0.05), and in the control group was 0.92 (SD = 0.05) (*p*-value = 0.012) (Figure 2).

Based on the median relative telomere length in the control group (0.915) (Table 3), the study subjects were categorized into two groups: ‘long telomeres’ (above the median) and ‘short telomeres’ (below the median). Statistically significant differences in the distribution of long and short telomeres between the studied groups were identified. There were 1.4 times more patients with short telomeres in the PE group compared to the control group (*p*-value = 0.044).

An analysis of the genotype and allele frequency distributions for SIRT1 rs12778366 and rs7895833 in relation to telomere length revealed no statistically significant differences between the studied groups (Appendix A).

Given the higher proportion of pregnant women with short telomeres in the PE group, further analysis focused on the distribution of genotypic and allelic frequencies specifically within the short telomere subgroup. Within the short telomere subgroup, the frequency of the C allele was significantly higher in the PE group (13.7%) compared to the control group (3.1%) (*p* = 0.027). The frequency of occurrence of heterozygous TC genotypes was higher in the PE group compared to the control group (27.5% and 6.2%, respectively, *p*-value = 0.019) (Table 4).

## 3. Discussion

This study investigated the relationship between the relative telomere length and two single nucleotide polymorphisms in the SIRT1 gene (rs12778366 and rs7895833) in pregnant women with GDM, exploring their potential role in the development of PE. Our findings indicate that women with PE and GDM had shorter relative telomere lengths compared to those with GDM alone. The odds of developing PE were 4.48 times higher in carriers of the heterozygous TC rs12778366 genotype under dominant and log-additive inheritance models (OR = 4.48; 95%CI 1.54–13.08; *p* = 0.003). Moreover, the PE group exhibited a higher prevalence of the heterozygous TC rs12778366 genotype among individuals with short telomeres.

Preeclampsia and GDM are the most common complications in pregnancy and are both influenced by genetic factors. The primary factor contributing to preeclampsia is the malfunction of vascular endothelial cells [47,48]. High levels of blood sugar, known as hyperglycemia, lead to chronic problems with blood clotting at the level of small blood vessels, known as microcirculation [49,50]. Pregnant women with both PE and GDM have a significantly higher risk of microvascular dysfunction compared to those with GDM only [51]. Damage to the endothelium can lead to shortened telomere length, which contributes to accelerated cell aging [52].

Several studies have demonstrated an association between shortened telomere length and an increased risk of pregnancy-related complications [53,54,55,56,57,58,59,60]. In addition, it has been reported that hypertension is associated with a shorter telomere length [53,54]. Telomere length during pregnancy has been studied mainly in the placenta, amnion, and umbilical cord blood [55,56,57,58]. The currently available literature data on the relationship of telomere length in peripheral blood leukocytes with preeclampsia are few and contradictory [59,60,61]. Evidence has supported the consistency of telomere length across different tissues; therefore, easily accessible leukocytes have been widely used as a substitute for tissue in measurements of telomere length [60]. Lekva et al. (2021) found no differences in telomere length when comparing control groups and women with PE [59]. A study by Zhang and colleagues (2022) showed that the relative telomere length of peripheral blood leukocytes and umbilical cord blood in patients with preeclampsia is longer than in women with a normal pregnancy [60]. However, Abu-Awwad et al. (2023) noted the shortening of peripheral blood leukocyte telomeres in patients with preeclampsia, which is consistent with our results [61].

There is evidence that SIRT1 may be involved in the development of preeclampsia. Studies have shown that SIRT1 deficiency can affect placental development and trophoblast invasion through autophagy [62,63,64,65,66,67,68,69,70,71]. SIRT1 can also protect endothelial cells from oxidative stress, inflammation, aging, and autophagy. It does this by deacetylating various substrates that may be involved in preeclampsia [62,63,64,65]. When SIRT1 is inhibited in hyperglycemic conditions, it leads to endothelial cell dysfunction. However, activating SIRT1 prevents the vascular cell damage caused by hyperglycemia [66,67].

According to some studies, the alternative alleles of rs12778366 and rs7895833 polymorphisms in the promoter region of the SIRT1 gene are associated with AMD, T2DM, obesity, glucose tolerance, hypertension development risk, and preeclampsia [38,39,40,41,42,43,44]. Figarska et al. (2013) reported that the SNP rs12778366 minor allele carriers had a better glucose tolerance [68]. GDM and T2DM share genetic polymorphisms, with the same effect size for the same risk alleles [69,70]. Han et al. (2015) mention that the C rs12778366 SNP allele had a positive correlation with a high susceptibility to T2DM [71]. Shimoyama et al. (2011) found that the A rs7895833 allele in females is associated with a high hypertension risk [43]. In a previous study, we identified the association of rs7895833 of the SIRT1 gene with the risk of developing preeclampsia in carriers of the genotype containing the SIRT1 G allele in a heterozygous state [44].

The main limitations of this study are the small sample size of the studied groups and the measurement of telomere length at only one point in time. The small size of the sample limits the generalizability of the results for a wider population. In addition, only patients with gestational diabetes were included in the study, which may have influenced the results. It is also likely that the risk factors for GDM and PE could have influenced the results. The study did not consider the risk factors that may affect telomere lengths, such as maternal age, pregestational BMI, gestational age, and level of glycemic control. In addition, the study did not consider the possibility of a causal inverse relationship, where short telomeres may be the result of GDM, obesity, or other somatic pathology prior to pregnancy rather than the cause of PE development. For future studies, it is important to consider that a larger sample size will provide more comparable data to the general population and increase the statistical relevance. Sequential measurements of RTL at different stages of pregnancy will allow for the establishment of cause–effect relationships and an estimation of the rate of body length depletion during pregnancy.

This study focused on pregnant women with GDM who are at increased risk of preeclampsia in a particular population. The study used appropriate statistical methods to analyze the data, ensuring that the results were reliable. The novelty of the research is also a strong point. As we did not find any data that estimate the correlation between relative telomere length and polymorphisms rs12778366 and rs7895833 in the SIRT1 in pregnancy complications, the results of our study can potentially serve as a basis for future research in this area.

## 4. Materials and Methods

The study involved DNA samples extracted from the whole venous blood of 124 pregnant women with GDM with gestational age greater or equal to 38 weeks who were followed up and gave birth in 2019/2022 in the Maternity Department of the State Clinical Hospital No. 29 (N.E. Bauman Hospital) of the Healthcare Department of Moscow. All respondents were Caucasian. Each participant gave written and informed consent in accordance with the Helsinki Declaration. The study was approved by the local Ethical Committee of the Perinatal Center at N.E. Baumann 29th Hospital, and the genetic part of the research was additionally approved by the Ethical Committee of the Research and Development Institute of General Pathology and Pathophysiology.

The diagnosis of GDM was established in accordance with the International Association of the Diabetes and Pregnancy Study Groups (IADPSG) recommendations and based on the criteria of the Russian National Consensus clinical guidelines “Gestational diabetes mellitus: diagnosis, treatment, postpartum care” [72,73]. Preeclampsia was diagnosed after 34 weeks (late-onset PE) based on the clinical guidelines “Hypertensive Disorders in Pregnancy, Labor and Post-Partum. Pre-eclampsia. Eclampsia” [11,12,13]. Pregnant women with a history of cardiovascular disease, any chronic disease, autoimmune disease, or inflammatory disease were excluded from the study.

The patients were divided into two groups: a study group consisting of patients with both GDM and PE (*n* = 61) and a control group consisting of pregnant women with GDM but without preeclampsia (*n* = 63).

### 4.1. Genomic DNA Extraction

Samples of the mothers’ whole venous blood were collected immediately before or during delivery. The blood samples were shipped by cold chain equipment to the laboratory and stored until analysis. The DNA was extracted from the whole venous blood and purified with the genomic DNA extraction kit (Evrogen LLC, Moscow, Russia) in accordance with the instructions of the manufacturer. The high-molecular DNA was stored at −20 °C. The quantity and quality of the isolated DNA was assessed in the NanoDrop 1000 spectrophotometer in accordance with accepted standards.

### 4.2. The Genotyping of rs12778366 and rs7895833 Polymorphisms in the SIRT1 Gene

The genotyping of the polymorphisms rs12778366 and rs7895833 of the SIRT1 gene in the promoter region was performed in real time using the technology of competing TaqMan probes according to the method taken from the literature. All primers and TaqMan probes were synthetically produced by Evrogen LLC, Russia (Table 5).

The reaction mixture for RT-PCR for one 20 μL sample contained 20 ng DNA, 70 mM Tris–HCl (pH 8.3), 2 mM ammonium sulfate, 0,02% BSA, 0.01% triton X-100, 0.01% sodium azide, pH 8.5–8.8, 125 mM dNTP, 200  μM forward primer, 200  μM reverse primer, 400  μM each of Taq-man probes, and 0.25 units of act. TaqDNA-polymerase.

The amplification was carried out in the CFX 96 programmable amplifier (Bio-Rad Laboratories, Inc., Hercules, CA, USA) with the subsequent thermocycling parameters for rs12778366 and rs7895833: initial denaturation for 5 min at 95 °C; then 40 cycles including denaturation at 95 °C for 30 s and at 60 °C for 30 s, with subsequent fluorescence pickup; and at 25 °C for 2 min at the end. To eliminate genotyping errors, 30% of the samples were randomly selected for re-genotyping and the results obtained were additionally evaluated.

### 4.3. Relative Telomere Length Assessment Using qPCR

The telomere length analysis was performed by real-time PCR on a CFX 96 programmable amplifier (Bio-Rad, USA) according to the original protocol taken from the literature (Cawthon, 2009) [74], using specific primers synthesized at Evrogen LLC, Russia (Table 6) [74].

The reaction mixture for the telomere analysis contained ~20 ng genomic DNA, 1× qPCRmix-HS SYBR (Evrogen LLC, Russia) telomere primer pair telg and telc (final concentrations 900 nM each), and beta-globin primer pair hbgu and hbgd (final concentrations 500 nM each) in a final volume of 25 µL. The thermal cycling profile was as follows: Stage 1—15 min at 95 °C; Stage 2—2 cycles of 15 s at 94 °C and 15 s at 49 °C; and Stage 3—32 cycles of 15 s at 94 °C, 10 s at 62 °C, 15 s at 74 °C with signal acquisition, 10 s at 84 °C, and 15 s at 88 °C with signal acquisition. The 74 °C reads provided the Ct values for the amplification of the telomere template, and the 88 °C reads provided the Ct values for the amplification of the beta-globin. For each sample and standard, there were three repeats of each telomeric and SCG reaction [74].

All obtained data were analyzed using CFX Manager TM software Version 3.01224.1015 (Bio-Rad). The relative length of the telomeres was estimated by the T/S index, which was calculated as the ratio of the number of copies of telomeric repeats to the number of copies of the reference gene. Samples of standard deviation (SD) of Ct ≥ 1 for telomeres or single copy gene (SCG) were excluded and were not included in the final study design.

### 4.4. Statistical Analysis

The statistical analysis was performed using SPSS 17.0 (SPSS, Chicago, IL, USA) and R 4.0.4 (R Foundation for Statistical Computing, Vienna, Austria). The normality of the data distribution was checked using the Shapiro–Wilk test. The data were presented as mean ± standard deviation (SD) and using absolute numbers with percentages. The Mann–Whitney test was used to compare the two groups. The distribution of long and short telomeres in the PE and control groups was compared using the χ^2^ test.

Testing for deviation from the Hardy–Weinberg equilibrium (HWE) was performed using the SNPassoc R package for both cases and controls separately before association analysis. It was considered that the SNP could be analyzed further if there were no significant deviations from the HWE in the control group (*p* > 0.05). Deviations from the HWE in the case group may have indicated issues such as the presence of an association of the analyzed polymorphism with the studied trait. Association analysis was used to evaluate the associations between SNP genotypes and alleles and PE risk by calculating the odds ratios (ORs) and their 95% confidence intervals (CIs) in SNPassoc R packages [75]. The selection of the best-fitting genetic model was based on the Akaike information criterion (AIC), whereby the best genetic models were those with the lowest AIC values. *p* < 0.05 was considered to indicate a statistically significant association.

## 5. Conclusions

Our investigation has revealed shorter telomeres in pregnant women with PE against a background of GDM compared to those who only have GDM. Moreover, the variant rs12778366 of the SIRT1 gene has been linked to both shorter telomeres and an increased risk of developing PE.

Despite the limitations that affect the generalizability and causal interpretation of these findings, the results underscore the need for further research. It is essential to consider the risk factors and consistently measure the RTL at various stages of pregnancy to validate and expand upon the results obtained in this study. Additional research is necessary to determine if there is a relationship between these markers and GDM or if they are exclusively associated with preeclampsia. This further investigation will help clarify the specific role of telomere length and genetic variants in the development of pregnancy-related complications and potentially lead to improved diagnostic and preventive strategies for both PE and GDM.

## Figures and Tables

**Figure 1 ijms-26-02967-f001:**
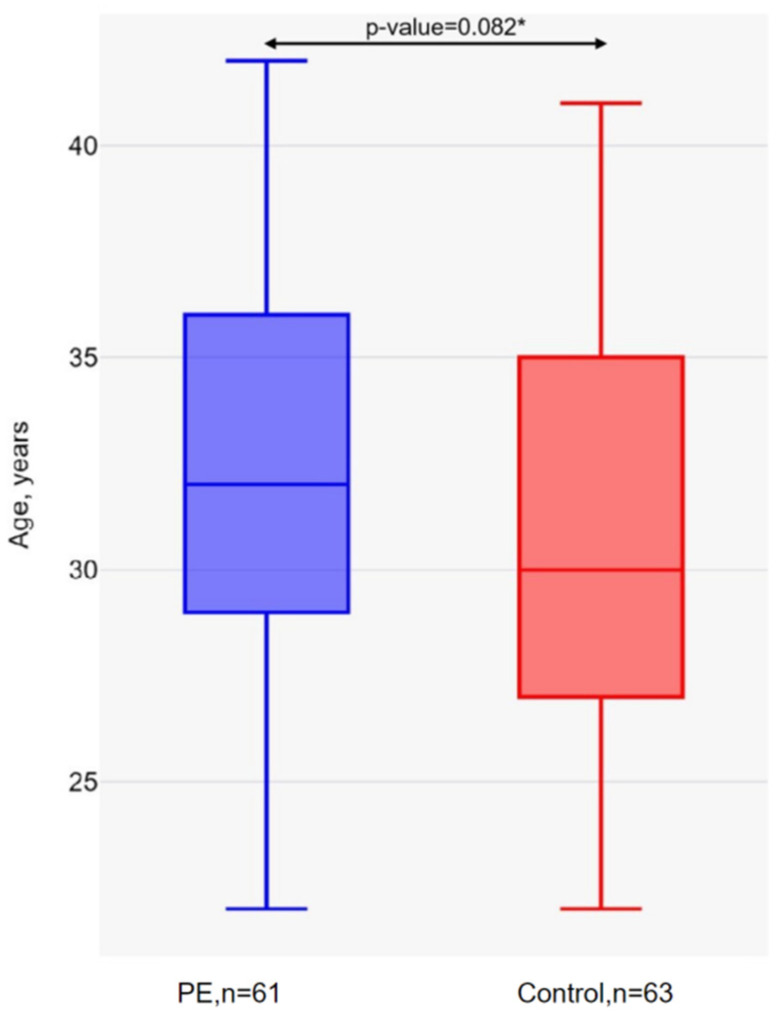
Age of pregnant women in the PE and control groups. * Mann–Whitney U-test.

**Figure 2 ijms-26-02967-f002:**
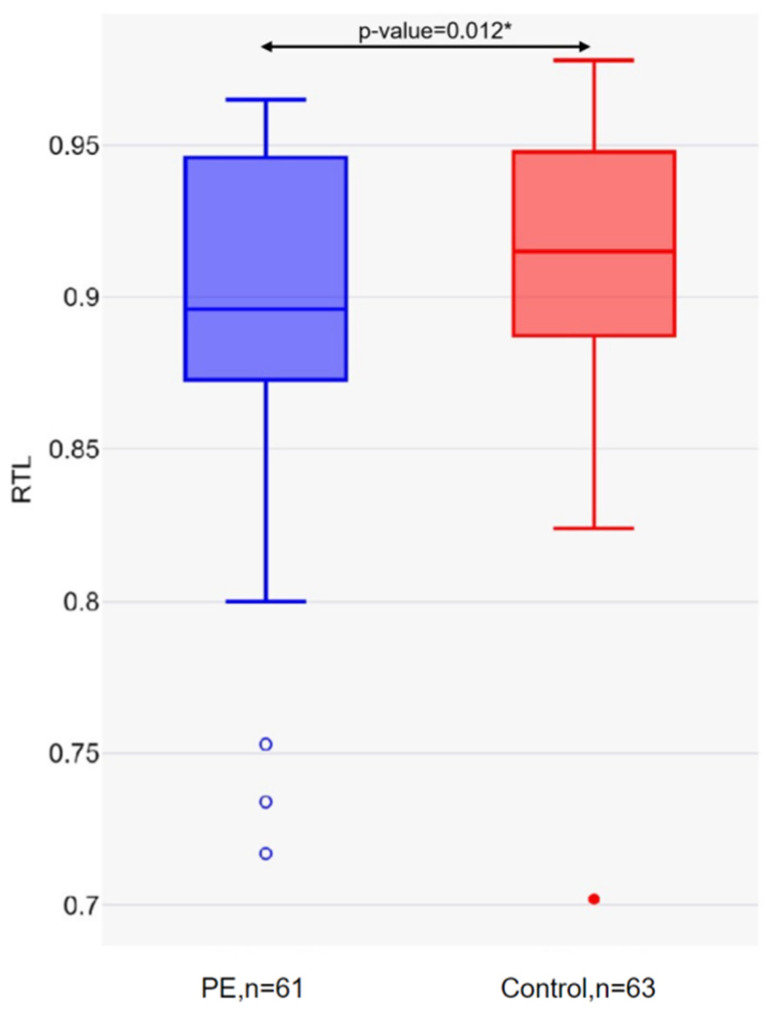
Relative leukocyte telomere length in the PE and control groups. * Mann–Whitney U-test.

**Table 1 ijms-26-02967-t001:** Analysis of the Hardy–Weinberg equilibrium for rs12778366 and rs7895833 of the SIRT1 gene in the control group.

SNP	Allele Frequencies		Genotype Distribution	*p*-Value
rs12778366	0.96 T	0.04 C	58/5/0	1
rs7895833	0.64 A	0.36 G	22/37/4	0.05

**Table 2 ijms-26-02967-t002:** Association of genotypes of polymorphisms rs12778366 and rs7895833 of the SIRT1 gene with PE in pregnant women with GDM.

SNP	Model of Inheritance	Genotypes	PE, *n* = 61	Control, *n* = 63	OR (95% of CI) ^1^	*p*-Value ^2^	AIC
rs12778366	Codominant	TT	44 (72)	58 (92)	1.00	** 0.003 **	167.1
		TC	17 (28)	5 (8)	4.48 (1.54–13.08)		
	log-additive	0, 1, 2	61 (49)	63 (51)	4.48 (1.54–13.08)	** 0.003 **	167.1
rs7895833	Codominant	AA	27 (44)	22 (35)	1.00	0.27	175.2
		AG	33 (54)	37 (59)	0.73 (0.35–1.51)		
		GG	1 (2)	4 (6)	0.20 (0.02–1.96)		
	Dominant	AA	27 (44)	22 (35)	1.00	0.29	174.7
		AG + GG	34 (56)	41 (65)	0.68 (0.33–1.39)		
	Recessive	AA + AG	60 (98)	59 (94)	1.00	0.17	174.0
		GG	1 (2)	4 (6)	0.25 (0.03–2.26)		
	Overdominant	AA + GG	28 (46)	26 (41)	1.00	0.6	175.6
		AG	33 (54)	37 (59)	0.83 (0.41–1.69)		
	log-additive	0, 1, 2	61 (49)	63 (51)	0.63 (0.33–1.20)	0.16	173.9

^1^ OR—odds ratio, CI—confidence interval; ^2^ Statistically significant results (*p*-value < 0.05) are highlighted in red and bold font.

**Table 3 ijms-26-02967-t003:** Distribution of telomeres by length in pregnant women with GDM in the PE and control groups.

Parameter	PE, *n* = 61	Control, *n* = 63	*p*-Value *
Long telomeres, *n* (%)	21 (34.43)	33 (52.38)	** 0.044 **
Short telomeres, *n* (%)	40 (65.57)	30 (47.62)	

* *p*-value is calculated using χ^2^. A statistically significant result (*p*-value < 0.05) is highlighted in red and bold font.

**Table 4 ijms-26-02967-t004:** Distribution of short telomeres depending on the genotypes rs12778366 and rs7895833 of the SIRT1 gene in pregnant women with GDM in the study groups (T/S median = 0.915).

SNP	Genotypes and Alleles	Short Telomere Ratio		*p*-Value *
		PE, *n* (%)	Control, *n* (%)	
rs12778366	TT	29 (72.5)	30 (93.8)	** 0.019 **
	TC	11 (27.5)	2 (6.2)	
	T	69 (86.3)	62 (96.9)	** 0.027 **
	C	11 (13.7)	2 (3.1)	
rs7895833	AA	15 (37.5)	8 (25.0)	0.528
	AG	24 (60.0)	23 (71.9)	
	GG	1 (2.3)	1 (3.1)	
	A	54 (67.5)	37 (57.8)	0.476
	G	26 (32.5)	23 (42.2)	

* *p*-value is calculated using χ^2^. Statistically significant results (*p*-value < 0.05) are highlighted in red and bold font.

**Table 5 ijms-26-02967-t005:** Sequences of oligonucleotides for RT-PCR of rs12778366 and rs7895833 of the SIRT1 gene.

SNP	Oligonucleotide Type	Sequences (5′ to 3′)
rs12778366	Forward primer Reverse primer Taq-Man probe for T allele Taq-Man probe for C allele	CCCCACGCAACCAAAGAT ATCGCTAAGGTCCTATCTACA FAM-TGGTCACCACT**A**TTCATTTCTGAA-BHQ1 HEX-TGGTCACCACT**G**TTCATTTCTGAA-BHQ1
rs7895833	Forward Revers Taq-Man probe for A allele Taq-Man probe for G allele	TTCTGAAGTAATGAGGTGG AGGAGACTCTGCCAGAAAT FAM-CCTACAGGAA**A**TCAACGTAA-BHQ1 HEX-CCTACAGGAA**G**TCAACGTAA-BHQ1

**Table 6 ijms-26-02967-t006:** Primer sequences for determining the relative telomere length.

Primer Name	Sequences (5′ to 3′)
telg	ACACTAAGGTTTGGGTTTGGGTTTGGGTTTGGGTTAGTGT
telc	TGTTAGGTATCCCTATCCCTATCCCTATCCCTATCCCTAACA
Hbgu	CGGCGGCGGGCGGCGCGGGCTGGGCGGcttcatccacgttcaccttg
Hbgd	GCCCGGCCCGCCGCGCCCGTCCCGCCGgaggagaagtctgccgtt

## Data Availability

The data that support the findings of this study are available on request from the corresponding author. The data are not publicly available due to privacy or ethical restrictions.

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
