# Peer review of "Increased Preeclampsia Risk in GDM Pregnancies: The Role of SIRT1 rs12778366 Polymorphism and Telomere Length"

_ijms, 2025, doi:10.3390/ijms26072967_

Round 1

Reviewer 1 Report

Comments and Suggestions for Authors

The authors try to explore the association between leukocyte telomere length and preeclamsia development in background of GDM. The telomere length has been investigated with agening researches extensively, yet in are of GDM and PE , dearth publications were found. The manuscript is well written and easy to follow however some points that are critical needs to be addressed before further progression.  

  1. I am wonder why the authors didn’t involve normotensive and euglycemic pregnant women as a control.
  2. Since the risk factors are many as authors mentioned, please provide at which trimester did you collect the blood samples of the measuring telomere ? As early preeclampsia is differed pathophysiological from late onset preeclampsia ! moreover, poor glycemic control can be a factor that provoke PE in GDM cases and also shorten the telomere. Accordingly ,can the authors provide adjusted odds ratio for maternal age, pre-gestational BMI and levels of glycemic control, gestational age.
  3. I Suggest to calculate the haplotype since the two variants were in the same gene The conclusion needs to be revised, biomarker for PE alone or PE+GDM?!
  4. Please explain how sample size was calculated
  5. Correct the odds ratio in the abstract
  6. Are there any unit used for telomere length !?
  7. In statistical analysis you mentioned performance of correlation analysis however, no results were shown!
  8. Line 149 is not clear revise

Author Response

We sincerely thank the reviewer for their valuable comment, which has helped improve the manuscript.

Below, we provide our response and the corresponding revision made.

Yours sincerely,

Comment 1. I am wonder why the authors didn’t involve normotensive and euglycemic pregnant women as a control.

Answer 1. In our study, we focused on women with gestational diabetes mellitus (GDM), as GDM is a separate independent risk factor for PE. Therefore, pregnant women with GDM were also included in the control group. 

Comment 2.  Since the risk factors are many as authors mentioned, please provide at which trimester did you collect the blood samples of the measuring telomere? 

Answer 2. Samples of whole venous blood of the mother were collected immediately before or during delivery (Line 234).

Comment 3. As early preeclampsia is differed pathophysiological from late onset preeclampsia ! 

Answer 3. The study included cases diagnosed after 34 weeks (late-onset PE). The text was corrected due to the reviewer comment (Line 225).

Comment 4. moreover, poor glycemic control can be a factor that provoke PE in GDM cases and also shorten the telomere. 

Answer 4. Glucose monitoring in pregnant women with GDM to assess carbohydrate metabolism compensation was conducted in accordance with the International Association of the Diabetes and Pregnancy Study Groups (IADPSG) recommendations and based on the criteria of the Russian National Consensus clinical guidelines "Gestational diabetes mellitus: diagnosis, treatment, postpartum care". Glucose targets did not differ in the study groups. 

In addition, RLT was measured at one point in time (the authors indicate this in Line 188-189), which does not allow us to assess the effect of glucose levels on changes in telomere length.

Comment 5. Accordingly, can the authors provide adjusted odds ratio for maternal age, pre-gestational BMI and levels of glycemic control, gestational age.

Answer 5.  In this study, the authors emphasize on the genetic aspects of this disease women with gestational diabetes mellitus and aims to investigate the correlation between telomere length and these selected SNPs. The factors you mention did not differ in the study groups (p-value > 0.05). Therefore, the authors decided not to include these variables in the logistic model.

In the discussion, the authors point out the weaknesses of the study, taking into account your comments:  The study did not consider risk factors that may affect telomere lengths, such as maternal age, pre-gestational BMI, gestational age and levels of glycemic control (Lines 194-195).

Comment 6. I Suggest to calculate the haplotype since the two variants were in the same gene 

Answer 6. We do not perform haplotype analysis because the sample size is small, which may limit statistical power and skew the results. In addition, rs12778366 is associated with the risk of developing preeclampsia, while rs7895833 does not have such an association, which indicates their independent inheritance. The low R2 value (0.078) between these SNPs confirms the absence of a functional link between them, despite the strong non-equilibrium coupling between these SNPs (D'=0.9818, p-value <0.0001). In this situation, it is impractical to conduct haplotype analysis. 

Comment 7. The conclusion needs to be revised, biomarker for PE alone or PE+GDM?!

Answer 7. The text was corrected due to the reviewer comment (Lines 298 - 310).

“Our investigation has revealed shorter telomeres in pregnant women with PE on the background of GDM compared to those who only have GDM. Moreover, the variant rs12778366 of the SIRT1 gene has been linked to both shorter telomeres and an increased risk of developing PE.

Despite the limitations that affect the generalizability and causal interpretation of these findings, they underscore the need for further research. It is essential to consider risk factors and consistently measure RTL at various stages of pregnancy to validate and expand upon the results obtained in this study. Additional research is necessary to determine if there is a relationship between these markers and GDM, or if they are exclusively associated with pre-eclampsia. This further investigation will help clarify the specific role of telomere length and genetic variants in the development of pregnancy-related complications and potentially lead to improved diagnostic and preventive strategies for both PE and GDM.”

Comment 8. Please explain how sample size was calculated

Answer 8. To obtain the minimum number of samples sufficient to conduct this study with sufficient statistical power (≥ 80%) at a significance level of 0.05 (with a two-sided alternative hypothesis), the quantitative calculation software QUANTO (Version 1.2.4, https://bio.tools/QUANTO), which takes into account the frequency of SNPs in the population and the prevalence of the disease. 

Comment 9. The odds ratio corrected in the abstract

Answer 9. - Corrected (Line 14)

Comment 10. Are there any unit used for telomere length !?

Answer 10. The relative length of telomeres was estimated by the T/S index. T/S describes the factor by which the sample differed from a reference DNA sample in its ratio of telomere repeat copy number to single copy gene copy number. The coefficient can be a number without units of measurement. 

Comment 11. In statistical analysis you mentioned performance of correlation analysis however, no results were shown!

Answer 11.  Line 291-292 deleted.

Comment 12. Line 149 is not clear revise

Answer 12.  Corrected (Line 152-153)

Reviewer 2 Report

Comments and Suggestions for Authors

Dear authors, thank you for your article. 

The relationship between telomere length and preeclampsia has been explored in previous studies; however, its role remains uncertain. In a large cohort study by Yang et al., which included 95 individuals with severe preeclampsia and 129 full-term healthy controls, no significant association was found between telomere length and preeclampsia. Instead, they identified interesting correlations between telomere length and factors such as gestational age and race. These variables should be considered in your study but do not appear to be addressed.

Additionally, most published studies on this topic have analyzed placental samples rather than blood samples, as used in your study. This methodological difference may present a potential limitation worth discussing.

Author Response

We are grateful for the reviewer's insightful feedback, which has greatly enhanced the quality of our manuscript. We have addressed their concerns and incorporated their suggestions into the revised version.

Sincerely,

Comment 1. The relationship between telomere length and preeclampsia has been explored in previous studies; however, its role remains uncertain. 

In a large cohort study by Yang et al., which included 95 individuals with severe preeclampsia and 129 full-term healthy controls, no significant association was found between telomere length and preeclampsia. Instead, they identified interesting correlations between telomere length and factors such as gestational age and race. These variables should be considered in your study but do not appear to be addressed.

Answer 1. In this study, the authors emphasize on the genetic aspects of this disease for women with gestational diabetes mellitus and aim to investigate the correlation between telomere length and these selected SNPs.

The study involved DNA samples of GDM pregnant women with gestational age greater or equal to 38 weeks (delivery gestational age in the PE group was comparable to delivery gestational age in the group without preeclampsia). All respondents were Caucasian, so we could not establish an association between  telomere length and race.

Corrections are performed in the text according to the reviewer comment.

Comment 2. Additionally, most published studies on this topic have analyzed placental samples rather than blood samples, as used in your study. This methodological difference may present a potential limitation worth discussing.

Answer 2. The authors used DNA samples extracted from the whole venous blood, because еvidence has supported that RTL in whole blood is an indicator of RTL in most tissues and the rate of telomere shortening is the same in different tissues [Demanelis et al. (2020); Dlouha et al. (2014); Daniali et al. (2013); Friedrich et al. (2000)].

Round 2

Reviewer 2 Report

Comments and Suggestions for Authors

Dear authors, thank you for correcting your article. I have no further comments. Good luck with your future work!

Author Response

Thank you